# Integrated Metabolomics and Transcriptomics Analysis of Flavonoid Biosynthesis Pathway in *Polygonatum cyrtonema* Hua

**DOI:** 10.3390/molecules29102248

**Published:** 2024-05-10

**Authors:** Luyun Yang, Qingwen Yang, Luping Zhang, Fengxiao Ren, Zhouyao Zhang, Qiaojun Jia

**Affiliations:** 1College of Life Sciences and Medicine, Zhejiang Sci-Tech University, Hangzhou 310018, China; 18957252806@163.com (L.Y.); qingwen_yang1013@163.com (Q.Y.); z2359224830@163.com (L.Z.); renfengxiao2022@163.com (F.R.); 13758266093@163.com (Z.Z.); 2Key Laboratory of Plant Secondary Metabolism and Regulation of Zhejiang Province, Zhejiang Sci-Tech University, Hangzhou 310018, China

**Keywords:** *Polygonatum cyrtonema* Hua, transcriptomics–metabolomics combined analysis, differential metabolites, flavonoid biosynthesis, qRT-PCR

## Abstract

Flavonoids, a class of phenolic compounds, are one of the main functional components and have a wide range of molecular structures and biological activities in *Polygonatum*. A few of them, including homoisoflavonoids, chalcones, isoflavones, and flavones, were identified in *Polygonatum* and displayed a wide range of powerful biological activities, such as anti-cancer, anti-viral, and blood sugar regulation. However, few studies have systematically been published on the flavonoid biosynthesis pathway in *Polygonatum cyrtonema* Hua. Therefore, in the present study, a combined transcriptome and metabolome analysis was performed on the leaf, stem, rhizome, and root tissues of *P. cyrtonema* to uncover the synthesis pathway of flavonoids and to identify key regulatory genes. Flavonoid-targeted metabolomics detected a total of 65 active substances from four different tissues, among which 49 substances were first study to identify in *Polygonatum*, and 38 substances were flavonoids. A total of 19 differentially accumulated metabolites (DAMs) (five flavonols, three flavones, two dihydrochalcones, two flavanones, one flavanol, five phenylpropanoids, and one coumarin) were finally screened by KEGG enrichment analysis. Transcriptome analysis indicated that a total of 222 unigenes encoding 28 enzymes were annotated into three flavonoid biosynthesis pathways, which were “phenylpropanoid biosynthesis”, “flavonoid biosynthesis”, and “flavone and flavonol biosynthesis”. The combined analysis of the metabolome and transcriptome revealed that 37 differentially expressed genes (DEGs) encoding 11 enzymes (C4H, PAL, 4CL, CHS, CHI, F3H, DFR, LAR, ANR, FNS, FLS) and 19 DAMs were more likely to be regulated in the flavonoid biosynthesis pathway. The expression of 11 DEGs was validated by qRT-PCR, resulting in good agreement with the RNA-Seq. Our studies provide a theoretical basis for further elucidating the flavonoid biosynthesis pathway in *Polygonatum*.

## 1. Introduction

*Polygonatum* is a medicinal food homology perennial plant in the family Asparagaceae [1,2]. Currently, three species of *Polygonatum* were listed in the Chinese Pharmacopoeia, namely *P. sibiricum* Delar. ex Redoute, *P. kingianum* Coll. et Hemsl, and *P. cyrtonema* Hua [3]. *Polygonatum* is neutral and sweet and has been used in China for over two thousand years. Modern pharmacological research has shown that it has various pharmacological activities, including anti-oxidant, hypoglycemic, hypolipidemic, anti-tumor, anti-bacterial, and anti-inflammatory and enhances immunity [4]. The chemical composition of *Polygonatum* includes polysaccharides, saponins, flavonoids, amino acids, inorganic salts, volatile oils, lignans, and alkaloids [5]. Among them, flavonoids are one of the main functional components.

Flavonoids are a large group of polyphenolic compounds in medical plants and exhibit a number of medicinal benefits, including anti-oxidation [6], anti-bacterial [7], and anti-tumor [8]. The flavonoids in *Engelhardia roxburghiana* Wall. can inhibit myocardial cell autophagy and lipid accumulation in thoracic aortas of high-fat-diet-treated mice [9]. *Polygonatum* flavonoids also exhibit similar beneficial effects [10,11,12,13]. In addition, different flavonoid compounds might demonstrate different pharmacological effects, such as homoisoflavonoids from *Polygonatum odoratum.* 3-(2′,4′-dihydroxybenzyl)-5,7-dihydroxy-6-methyl-chroman-4-one could inhibit the proliferation of cancer cells [14]. Tectoridin shows the effects of sensitizing adipocytes for insulin, which promotes the full utilization of insulin and lowers blood glucose concentrations [15]. Odoratumone A exhibits stronger DPPH radical scavenging activities [16]. The homoisoflavonoids 5,7-dihydroxy-3-(2-hydroxy-4-methoxybenzyl)-8-methyl-chroman-4-one isolated from *Polygonum verticillatum* have a considerable antibacterial action [17]. Considering the differences in the efficacy of flavonoids in *Polygonatum*, it is necessary to separate and identify the monomer of such compounds, which will provide a more rigorous and systematic basis for clinical applications.

Since 1969, three flavonoids have been isolated from the fresh leaves of *Polygonatum multiflorum* on the basis of chemical, UV, and H-NMR spectral properties [18]. Using various chromatography and HPLC techniques for extraction and purification, 90 flavonoids were gradually identified from the genus *Polygonatum* based on their physicochemical properties and spectral characteristics. Metabolomics technology employs multi-reaction monitoring techniques to collect vast amounts of data and establish a metabolite specimen database, achieving high-throughput and accurate qualitative and quantitative analysis of metabolites [19], and providing more comprehensive and precise information on metabolites [20]. In recent years, 53 flavonoids have been isolated from the genus *Polygonatum* using metabolomics techniques. UPLC-ESI-MS/MS analysis revealed 22 flavonoids in the rhizomes of *P. cyrtonema*, including chalcones, dihydroflavones, dihydroflavonols, flavones, flavonoids, flavonoid carbonosides, and flavonols [21]. Sharma et al. [22] employed UPLC-MS/MS to identify 10, 15, and 2 flavonoids in *Polygonatum verticillatum*’s fruits, leaves, and rhizomes, respectively. Research demonstrated that leaf flavonoid content was the highest in all tested tissues of *P. sibiricum* in the forest [23]. Roots, leaves, flowers, and fruits of *Polygonatum* were edible, which were recorded in the Collection of Commentaries on the Classic of the Materia Medica (Ben Cao Jing Ji Zhu) compiled by Tao Hongjing in the Southern Liang dynasty (502–557 AD). Most of the *Polygonatum* flavonoids were isolated from rhizomes [24]. Yet, few studies have been published about flavonoids from the other tissues of *Polygonatum*. Therefore, it is necessary to study the flavonoids in different tissues of *Polygonatum*.

Medicinal plants contain an abundance of secondary metabolites, which are controlled by a cluster of functional genes. RNA-Seq technology, which utilizes high-throughput sequencing to analyze the transcriptome of organisms, has been widely applied in new gene identification, molecular markers development, and metabolic pathways elucidation in medicinal plants [25,26,27,28]. As a result, transcriptomics is an effective way to identify genes involved in secondary metabolite synthesis. The biosynthesis of flavonoids is regulated by a series of enzymes related to the phenylpropanoid and flavonoid pathways in medicinal plants [29]. Phenylalanine ammonialyase (PAL) is the first enzyme of the phenylpropanoid pathway, and chalcone synthase (CHS) is the entry-step enzyme that is responsible for the biosynthesis of the flavonoid backbone [30]. Chalcone isomerase (CHI) catalyzes the formation of dihydroflavonoid compounds, which then undergo intramolecular cyclization reactions via the action of dihydroflavonol 4-reductase (DFR) and flavonol synthase (FLS) on different branches, resulting in the evolution of various flavonoid compounds such as flavones, flavonols, isoflavones, and anthocyanins [31]. Transcription analysis obtained a total of 18, 15, and 26 enzymes related to the flavonoid biosynthesis pathway in *Ephedra sinica* Stapf [32], *Scutellaria barbata* D. Don [33], and *Sophora japonica* Linn [34], respectively. In *Ginkgo biloba*, 13 gene families encoding 111 enzymes involved in flavonoid biosynthesis were identified and characterized [35]. In *Polygonatum*, a total of 26 key enzymes associated with flavonoid synthesis pathways were identified through transcriptional analysis [36,37]. Therefore, such a pathway in *Polygonatum* still remains unclear and requires further exploration.

The integration analysis of metabolomics with transcriptomics can elucidate the relationship between genes and metabolites more effectively, so as to provide a more accurate and reliable result [38]. Integrative transcriptional and metabolic analysis of peanut skins with significant differences in anthocyanin types and contents showed that flavonoid biosynthesis was the key pathway for the formation of seed coat color, and delphinidin and cyanidin were the main differential metabolites [39]. Through comprehensive analysis of metabolomics and transcriptomics, Wang et al. [40] found that terpenes were the main volatile compounds, and 17 key genes were significantly related with the terpenoid synthesis pathway, providing references for understanding the aroma biosynthesis and perfume formulations of *Dendrobium loddigesii* Rolfe. This study aims to identify flavonoid compounds in four tissues of *Polygonatum*, to determine the flavonoid synthesis pathway and its regulatory mechanisms through integrated metabolomics and transcriptomics techniques. The results of the present study will provide new insights for investigating the transcriptional regulation of flavonoids in *Polygonatum*.

## 2. Results

### 2.1. Metabolic Profiling

The phenolics metabolites of the four tissues from *P. cyrtonema* were investigated based on UPLC-ESI-MS/MS. A total of 65 active substances were identified, among which 49 substances were first study to identify in *Polygonatum* (Appendix A), and 38 substances were flavonoids, including 14 flavonols, 9 flavones, 7 flavanones, 3 dihydrochalcones, 2 anthocyanins, 2 isoflavones, and 1 flavanol (Table 1). The highest number and content of metabolites were detected in leaves and the lowest in rhizomes. There were 17 metabolites shared across the tissues, among which rutin had the highest proportion of flavonoid monomers in each tissue. The top ten metabolites with the highest content are shown in Appendix A, which belong to flavonols, flavones, benzoic acid derivatives, and phenylpropanoids (Appendix A). 

### 2.2. Metabolite Analysis Based on PCA and OPLS-DA

Principle component analysis (PCA) indicated that the replicates for each tissue grouped was based on the detected metabolites, suggesting that the results were reliable, and there was less degree of variability within the tissues (Figure 1). In this study, a supervised orthogonal partial least-squares discriminant analysis (OPLS-DA) model was employed to compare the metabolites content of the samples in pairs to evaluate the difference between Leaf and Stem (R^2^X = 0.987, R^2^Y = 0.998, Q^2^ = 0.997), Leaf and Rhizome (R^2^X = 0.996, R^2^Y = 1, Q^2^ = 0.999), Leaf and Root (R^2^X = 0.995, R^2^Y = 1, Q^2^ = 0.999), Stem and Rhizome (R^2^X = 0.998, R^2^Y = 1, Q^2^ = 0.99), Stem and Root (R^2^X = 0.995, R^2^Y = 1, Q^2^ = 0.99), and Rhizome and Root (R^2^X = 0.965, R^2^Y = 0.997, Q^2^ = 0.991) (Figure 2a–f). The Q^2^ values of all comparison groups exceeded 0.9, demonstrating that such models were stable and reliable and could be used to further screen differential flavonoid metabolites.

### 2.3. Differentially Accumulated Metabolite Screening in Leaf, Stem, Rhizome, and Root Tissues

Pairwise comparisons were conducted among the six groups and a total of 62 differentially accumulated metabolites (DAMs) were screened out. Among all comparison groups, the leaf and rhizome had the most DAMs, while the rhizome and root showed the least (Figure 3). Moreover, most metabolites in leaves were upregulated. After taking the intersection of each comparison group, isorhamnetin, isorhamnetin-3-O-glucoside, malvin, vitexin, and phlorizin were common DAMs in the six comparison groups, while myricetin and 4-hydroxycinnamic acid were only screened between rhizomes and leaves, as well as between rhizomes and stems, respectively (Appendix A).

### 2.4. Functional Annotation and Enrichment Analysis of Differentially Accumulated Metabolites

To obtain detailed pathway information, DAMs of each comparison group were annotated by searching against the Kyoto Encyclopedia of Genes and Genomes (KEGG) database (Appendix A). The results showed that DAMs from each comparison were enriched in flavonoid biosynthesis pathways (Figure 4), including phenylpropanoid biosynthesis (ko00940), flavonoid biosynthesis (ko00941), and flavone and flavonol biosynthesis (ko00944). A total of 19 DAMs were annotated into these pathways, including 5 flavonols, 3 flavones, 2 dihydrochalcones, 2 flavanones, 1 flavanol, 5 phenylpropanoids and 1 coumarin (Table 2).

### 2.5. Functional Annotation and Enrichment Analysis of Differential Flavonoid Genes

RNA-sequencing analysis was performed on the leaf, stem, rhizome, and root of *P. cyrtonema*. The differentially expressed genes (DEGs) identified from the six pairs of comparisons were further subjected to KEGG pathway enrichment analyses to screen genes associated with flavonoid biosynthesis (Appendix A). The results showed that a total of 222 unigenes encoding 28 enzymes were annotated into flavonoid synthesis pathways, and 7 enzymes were first identified in *Polygonatum* (Table 3). Most of the DEGs were annotated into the phenylpropanoid biosynthetic pathway (ko00940); 47 DEGs were annotated to the biosynthesis of flavonoid (ko00941); the flavone and flavonol biosynthesis pathway was associated with 6 annotated DEGs (ko00944) (Table 3). 

### 2.6. Combined Analysis of Transcriptome and Metabolome Analysis

The Pearson correlation coefficient was used to calculate the relationship between gene expression and metabolite level. The results revealed that 37 DEGs encoding 11 key enzymes were significantly correlated with 19 DAMs (Figure 5a). In addition, the expression profiles of 6 genes (*4CL1*, *4CL2*, *CHS2*, *CHS3*, *DFR1*, *LAR*) correlate well with those of the flavonoid abundance, which was significantly positively correlated (*p* < 0.01) with the content of astragalin, catechin, chlorogenic acid, cosmosiin, ferulic acid, kaempferol, naringin, phlorizin, quercetin, and quercitrin. Furthermore, significant positive associations were discovered between luteolin and *4CL2*, *CHS3*, phloretin and *4CL1*, *4CL2*, *CHS3*, rutin and *4CL1*, *4CL2*, *CHS3*, *DFR1*, *LAR*, and between sinapic acid and *4CL1*, *CHS2*, *DFR1*, *LAR.* It seemed that these 14 metabolites were more likely to be regulated in the flavonoid biosynthesis pathway, which was in correspondence with the results of the heatmap (Figure 5). Due to the KEGG pathway, transcriptomic and metabolomic data, we outlined potential pathways for flavonoid biosynthesis and accumulation in *P. cyrtonema* (Figure 5b). 

### 2.7. Validation of Differentially Expressed Genes by Quantitative Real-Time PCR

To validate the accuracy of the RNA-Seq data, 11 differentially expressed genes encoding 9 key enzymes were selected for a quantitative real-time PCR (qRT-PCR) analysis. As illustrated in Figure 6, the expression patterns of these genes were consistent with the corresponding FPKM values derived from RNA-Seq analysis, which indicated the reliability of the transcriptome. The core genes for flavonoid biosynthesis, such as *PAL3*, *C4H2*, *CHS1*, *CHS4*, *F3H*, *DFR1*, *LAR*, and *ANR*, were more highly expressed in the leaves than those in the rhizome.

## 3. Discussion

Mass spectrometry-based metabolomics approaches allow for detection and quantification of many thousands of metabolic products simultaneously, shedding light on the metabolic pathways of effective chemical synthesis in medicinal plants. In the present study, a wide-ranging analysis of phenolic metabolites using targeted metabolomics was conducted in four different tissues of *P. cyrtonema*, and a total of 65 phenolic metabolites from 12 categories were detected (Appendix A). Flavones and flavonols were the main phenolic metabolites. The highest proportion of flavones and flavonols in flavonoids was also reported in *Tetratigma hemsleyanum* Diels et Gilg [42], *Hippophae rhamnoides* L. [43], Ziziphi Spinosae Semen [44], *Kadsura coccine*a [45], and *Cannabis sativa* L. [46]. Moreover, the content of flavonoids was highest in leaves and lowest in rhizomes (Table 1), which was also reported in *P. sibiricum* in the forest [23]. The higher content of flavonoids in the leaves was also reported in *P. cyrtonema* [47], *Bupleurum chinense* DC. [48], *Perilla frutescens* (L.) Britt. [49], *Hippophae rhamnoides* L. [50], and *Dendrobium officinale Kimura et* Migo [51]. It seemed that leaves were the main source of flavonoids, which could be related to chemical defense in plants. Flavonoids are phenolic compounds that have an undesirable bitter taste, which helps to reduce herbivory [47]. In addition, flavonoids are the primary UV absorbers, preventing or limiting UV-B damage to plants [48]. The most abundant flavonoid in *Polygonatum* was rutin, which belongs to the flavonols. Such result was also described in Golden Buckwheat and *Lonicera japonica* Thunb. [52,53]. Among the detected active substances, 49 substances were first study to identify in this species (Appendix A). Researchers have extracted a total of 127 *Polygonatum* flavonoids, which are categorized according to their chemical structures into the classes of homoisoflavones, isoflavones, flavonoids, chalcones, and dihydroflavonoids. Homoisoflavones, a special subclass of flavonoids [54], are rarely found in nature, mainly existing in Fabaceae and Asparagaceae families [55], and accounted for nearly half of the total identified flavonoid compounds. Only 19 substances of the identified *Polygonatum* flavonoids had biological reference standards and were investigated with UPLC-ESI-MS/MS in the four tissues of *P. cyrtonema*. Fifteen of them were successfully detected, indicating high detection efficiency of such metabolomics (Appendix A). Three of the remaining four compounds, isoliquiritigenin, liquiritigenin, and jaceosidin, were exclusively detected in *P. kingianum* [24,56,57]. However, apigenin was not detected in this study, possibly due to its low concentration. The top ten metabolites listed in Appendix A had the effects of anti-oxidation and anti-bacterial [58,59,60,61,62,63,64,65,66,67]. In addition, most of them also have anti-tumor (except salicylic acid), and hypoglycemic (except salicylic acid and vanillin) effects.

In the current study, both DAMs and DEGs annotated by the KEGG database were mainly enriched in phenylpropanoid biosynthesis (ko00940), flavonoid biosynthesis (ko00941), and flavone and flavonol biosynthesis (ko00944) pathways (Table 2 and Appendix A). In medicinal plants, differential genes related to flavonoid biosynthesis are subjected to the phenylpropane biosynthesis pathway [68,69,70,71], flavonoid biosynthesis pathway [69,71], flavone and flavonol biosynthesis pathways [69,70,71], and isoflavone biosynthesis pathway [69,71] by KEGG pathway analysis. The first three pathways have also been reported in *P. kingianum* [41] and *P. cyrtonema* [36]. After comparing with other results in *Polygonatum*, our study identified more genes that were 194, 47, and 6 genes in the phenylpropane biosynthesis, flavonoid biosynthesis, flavonoid and flavonol biosynthesis pathways, respectively. Significantly different phenolic compounds contents in the detected tissues helps to identify more DEGs. The transcriptome analysis identified 28 enzymes encoded by flavonoid-related DEGs, of which 21 were reported in *Polygonatum*, including *CHI*, *CHS*, *CCOMT*, and *CYP73A* [36,37,41,72] (Table 3). Among the 7 newly identified enzymes, *katG*, *TOGT1*, *CSE*, and *PRDX6* were enriched in phenylpropanoid biosynthesis, *PGT1* and *FNS* were annotated to flavonoid biosynthesis, and *FG2* was annotated to flavone and flavonol biosynthesis. Flavone synthase (FNS) catalyzes the synthesis of flavones from dihydroflavones. *LjFNS* was first cloned from *Lonicera japonica* using Chromosome walking and might be involved in flowers development due to the highest expression in flowers by qRT-PCR analysis [73]. Integrated metabolomic and transcriptomic study revealed that both upregulated DAMs and DGEs were associated with the flavonoid biosynthesis pathway, and the upregulation of the expression of genes phlorizin synthase (*PGT1*), shikimate O-hydroxycinnamoyltransferase (*HCT*), and chalcone synthase (*CHS*) promotes p-coumaroyl-CoA in sugarcane roots transferred into homoeriodictyol chalcone and 5-deoxyleucopelargonidin [74]. The partial cDNA of flavonol-3-O-glucoside L-rhamnosyltransferase (*FG2*) involved in the rutin biosynthetic pathway was first cloned in *C. spinosa* plants [75]. Other medicinal plants such as soybeans [76], sugarcane [77], and *Populus alba×P*. glandulosa also successfully isolated *FG2* [78]. The constructed flavonoid synthesis pathway and the related newly identified genes provide a reference for subsequent functional studies of the candidate genes and comparative studies with other species. 

The association analysis of metabolomics and transcriptomics confirmed that 19 DAMs regulated in the flavonoid biosynthetic pathway were significantly associated with 37 DEGs encoding 11 enzymes (Figure 5a). Through de novo transcriptome assembly and metabolomic analysis of shell and leaves of *Trapa bispinosa* Roxb, both the 15 DEGs (*CHS*, *FLS*, *DFR*, *ANR*…) and 42 metabolites (luteolin, delphinidin…) involving phenylpropanoid and flavonoid biosynthesis were identified to construct flavonoid biosynthesis pathways and related networks (Yin et al.) [79]. Combined transcriptome and metabonomic data of *C. paliurus* leaves at different developmental stages revealed that the expression trends in differential flavonoids metabolites and genes were significantly related [80]. Among DEGs, six genes (*4CL1*, *4CL2*, *CHS2*, *CHS3*, *DFR1*, *LAR*)-related flavonoid biosynthesis also showed strong correlations with DAMs (Figure 5a). Coumaric acid generates coumaroyl-CoA in the presence of 4CL, which was the primary rate-limiting enzyme in the phenylalanine pathway [81]. Both *4CL1* and *4CL2* were strongly positively linked with 14 DMFs, including catechin, kaempferol, and caffeic acid, etc., which was consistent with results in *Lycium chinense* fruits [82]. CHS was the first and major rate-limiting enzyme in the flavonoid biosynthesis pathway and was one of the most explored enzymes in medicinal plants. It catalyzed the formation of chalcone, a substance from which different flavonoids were derived [83]. The overexpression of *EaCHS1* in peels of C. *reticulata* ‘Chachi’ increased the production of downstream flavonoids and the expression of related genes in the phenylpropanoid pathway [84]. There was a substantial positive association between *SeCHS1* and five flavonoid compounds in *Sechium edulel* [85]. Our findings also revealed that *CHS2* and *CHS3* showed a highly positive relationship with 12 and 13 differential metabolites, respectively (Figure 5a). In addition, a negative association was found between *CHS2* and umbelliferone. Flavonoid content was negatively correlated with the expression of *CHS* during fruit development of *Lycium chinense* [82]. This may be due to spatiotemporal expression differences or functional redundancy of *CHS* genes. DFR was an essential enzyme responsible for the diversification of anthocyanins and ellagitannins [86]. The expression of the *DFR1* gene was highly positively correlated with the content of 12 flavonoid metabolites in this study. *DFR* gene expression positively correlated with the accumulation of flavonoids in leaves of field conifers under heavy metal stress [87]. Wang et al. [88] found a strong positive correlation of the accumulation of catechin and eriodictyol with the *DFR* gene expression in red cell line of *Saussurea medusa*. LAR was a key enzyme in the plant flavonoid synthesis pathway that catalyzed the conversion of colorless anthocyanins into catechins [89]. The expression of the *LAR* gene was positively associated with 12 flavonoids in the current investigation, except for umbelliferone. Correlation analysis revealed that the content of total flavonoids, total flavanols, and four flavanol components including catechins and epicatechin showed a significantly strong correlation with the *LAR* gene in jujube fruits [90].

We also demonstrated that the majority of genes involved in the synthesis pathway of flavonoids exhibited heightened expression levels in the leaves and displayed lower expression levels in rhizomes, which was further confirmed by qRT-PCR (Figure 6). As a result, the most differential metabolites were identified between leaf and rhizome (Figure 3). In general, these preliminary results demonstrated the correlation of metabolites and genes in the flavonoid synthesis pathway and established the foundation for future research on the specific mechanisms regulating the synthesis of secondary metabolites in *P. cyrtonema*.

## 4. Materials and Methods

### 4.1. Plant Materials

*Polygonatum cyrtonema* Hua, as a wild resource, was collected in Qingyang, Anhui Province (30°34′15″ N, 117°48′22″ E), and then planted in Yuhang, Zhejiang Province (30°48′56″ N, 119°76′04″ E), in November 2018. The samples were identified by Zongsuo Liang, professor at the Zhejiang Sci-Tech University. The root, rhizome, stem, and leaf of *P. cyrtonema* were collected from three individual plants on the sunny day of 22 April 2022. The biological replicates were conducted three times for each tissue. All samples were cleaned with ultrapure water and were immediately frozen in liquid nitrogen and stored at −80 °C for metabolomic, transcriptomic analyses and quantitative real-time PCR analysis.

### 4.2. Metabolome Analysis Based on UPLC-ESI-MS/MS

Samples were crushed for 2 min at 60 Hz using a grinder and then dissolved in a methanol–water solution (V (water):V (methanol) = 1:2) before metabolic analysis. To improve the extraction rate, we conducted ultrasound three times and vortexed twice. After centrifugation of the liquid (10,000 rpm, 10 min), the supernatant was aspirated, filtered through a microporous membrane (0.22 μm pore size), and stored in brown injection vials for UPLC-ESI-MS/MS analysis.

Liquid chromatography was performed using a Waters ACQUITY UPLC HSS T3 (Milford, MA, USA, 1.8 μm, 2.1 mm × 100 mm) column. The mobile phase consisted of 0.1% formic acid aqueous solution for phase A and acetonitrile for phase B. The mass spectrometry conditions used in this study involved Ionspray voltage (IS) at 5500 V or −4500 V, curtain gas (CUR) at 35 psi, and a Source Temperature at 500 °C. To perform absolute quantitative analysis on 130 phenolic compounds (Appendix A), we employed a phenolic compound analysis method package based on the LC-MS/MS platform developed by LuMing Biotechnology Co., Ltd. (Shanghai, China). Targeted metabolites were analyzed in Schedule multiple reaction monitoring (SRM) mode. The MRM pairs, declustering potentials (DP), and collision energies (CE) were optimized for each analyte. Data acquisitions and further analysis were conducted using Analyst software (https://sciex.com/products/software/analyst-software). SCIEX OS-MQ software (https://sciex.com/products/software/sciex-os-software) was used to quantify all metabolites. The QCs were injected at regular intervals (every 10 samples) throughout the analytical run to provide a set of data from which repeatability can be assessed. The data analysis was based on peak area normalization. The detection parameters of reference substance were deselected for peak retention time alignment with minimum intensity at 15% of base peak intensity, noise elimination level at 10.00, and isotopic peaks were excluded.

### 4.3. Transcriptomic Analysis

Total RNA was extracted using the mirVana miRNA Isolation Kit, and RNA integrity was evaluated using the Agilent 2100 Bioanalyzer (Agilent Technologies, Santa Clara, CA, USA). The libraries were constructed using the TruSeq Stranded mRNA LTSample Prep Kit (Illumina, San Diego, CA, USA), according to the manufacturer’s instructions. Then, these libraries were sequenced on the Illumina sequencing platform (HiSeqTM 2500 or Illumina HiSeq X Ten), and 125 bp/150 bp paired-end reads were generated.

Raw data (raw reads) were processed using Trimmomatic [91] (http://www.usadellab.org/cms/index.php). The reads containing ploy-N and the low-quality (Q < 20) reads were filtered out to obtain the clean reads. The clean reads were de novo assembled into transcripts by using Trinity [92] (vesion:2.4) in the paired-end method. The longest transcript of each unigene was chosen for subsequent analysis.

### 4.4. Quantitative Real-Time PCR (qRT-PCR) Analysis

Quantitative real-time PCR (qRT-PCR) was performed using the Taq Pro Universal SYBR qPCR Master Mix Kit (Vazyme, Nanjing, China). Gene primers designed using Primer 6.0 are listed in Table 4. The 10 µL reaction mixture contained 1 µL of cDNA, 0.2 µL of each primer, 5 µL of 2× SYBR qPCR Master Mix, and 3.6 µL of RNase-free water. All qRT-PCR analyses were performed using the following conditions: denaturation at 95 °C for 30 s, followed by 40 cycles of 95 °C for 10 s, and then at 60 °C for 30 s. All reactions were repeated three times in the experiments, and the 2^−ΔΔCt^ method [93] was used to calculate the relative expression of each unigene.

### 4.5. Data Analysis

Principal component analysis (PCA) and orthogonal partial least-squares discriminant analysis (OPLS-DA) were used to analyze the trend of metabolites. The Student’s *t*-test was performed for pairwise comparisons. The differentially accumulated metabolites and expressed genes were screened according to *p* ≤ 0.05 and |log2 (fold change)| > 1 rules. Based on the Kyoto Encyclopedia of Genes and Genomes (KEGG) database, the related pathways of the DAMs and DEGs were determined. The OECloud tools (https://cloud.oebiotech.com) (accessed on 20 March 2023) application was used for Pearson’s correlation analysis. A significance level of *p*  <  0.05 was considered statistically significant.

## 5. Conclusions

In the present study, metabolomics and transcriptomics were employed to identify metabolites and genes associated with the flavonoids biosynthesis pathway, respectively. Most of the detected substances were first study to identify in *Polygonatum.* A total of 222 differentially expressed genes encoding 28 enzymes and 14 different flavonoid metabolites were identified in four tissues of *P. cyrtonema*. Integrated analysis of transcriptomic and metabolomic data revealed that these 14 different flavonoid metabolites were more likely to be regulated by flavonoid biosynthesis-related genes, especially *4CL1*, *4CL2*, *CHS2*, *CHS3*, *DFR1*, and *LAR*. Our results provide valuable information on *Polygonatum* flavonoid regulation.

## Figures and Tables

**Figure 1 molecules-29-02248-f001:**
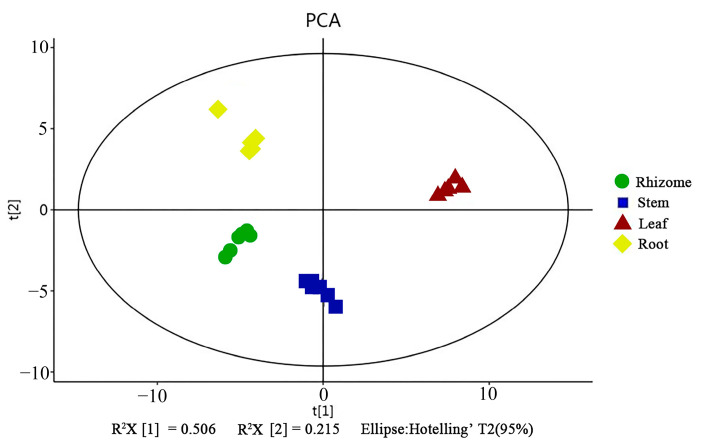
Metabolite analysis on the basis of principal component analysis (PCA).

**Figure 2 molecules-29-02248-f002:**
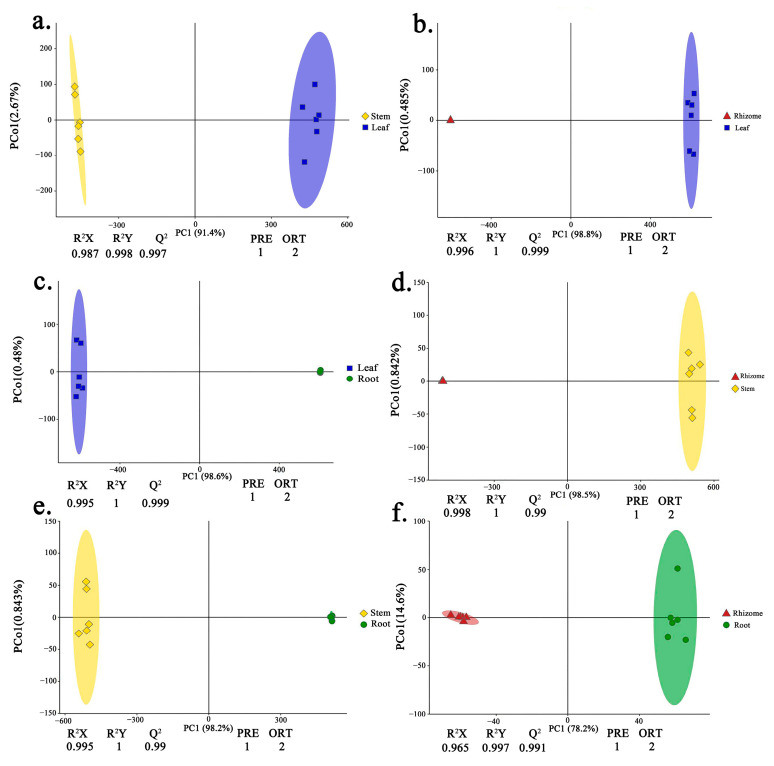
Metabolite analysis based on orthogonal partial least-squares discriminant analysis (OPLS-DA). (**a**–**f**) OPLS-DA model plots for the comparison group Leaf vs. Stem, Leaf vs. Rhizome, Leaf vs. Root, Stem vs. Rhizome, Stem vs. Root, Rhizome vs. Root, respectively. The colorful shading represents the confidence ellipse.

**Figure 3 molecules-29-02248-f003:**
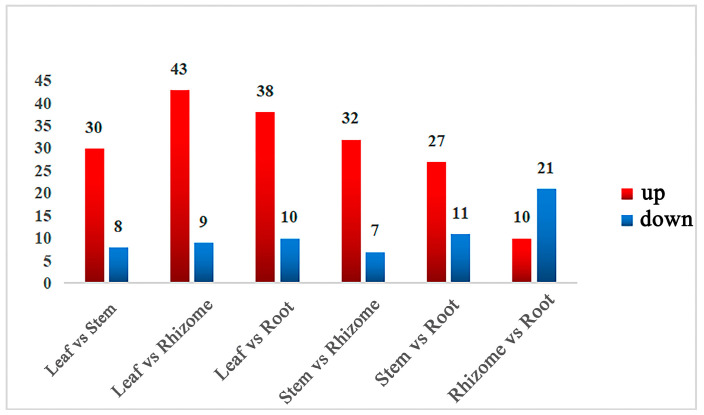
Statistics of differently expressed metabolites. The statistical analysis of the number of differential metabolites for all different groups is presented, with the red color indicating upregulated metabolites and the blue color indicating downregulated metabolites.

**Figure 4 molecules-29-02248-f004:**
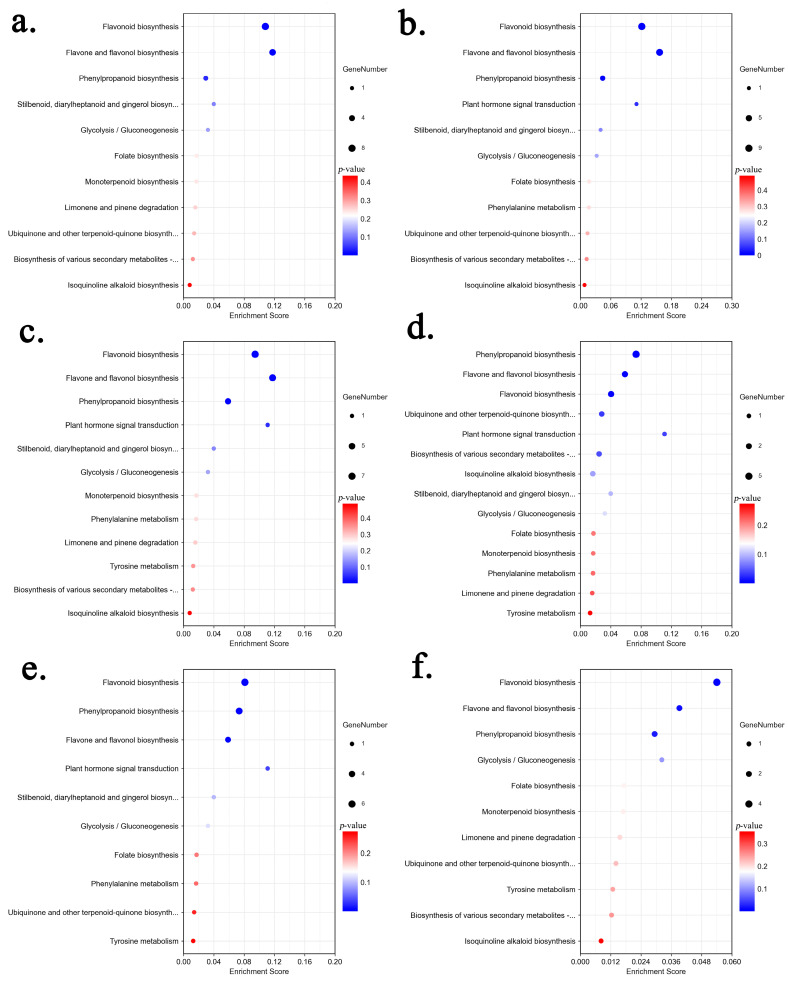
KEGG pathway enrichment of the differential metabolites in each comparison: (**a**) Leaf vs. Stem, (**b**) Leaf vs. Rhizome, (**c**) Leaf vs. Root, (**d**) Stem vs. Rhizome, (**e**) Stem vs. Root, (**f**) Rhizome vs. Root.

**Figure 5 molecules-29-02248-f005:**
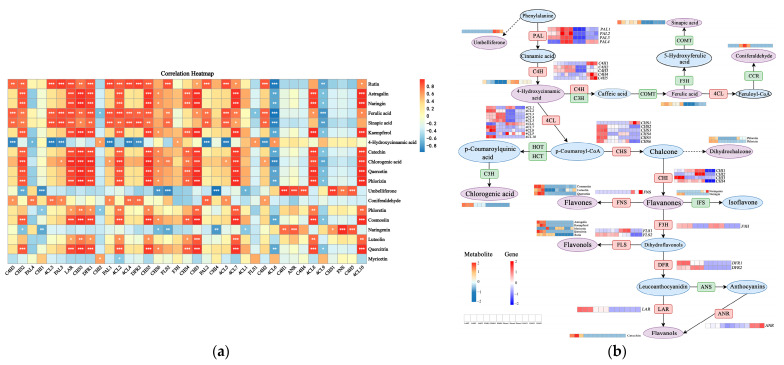
Transcription metabolism conjoint analysis data plot. (**a**) Correlation heatmap between differentially accumulated metabolites and differentially expressed genes. * *p* < 0.05, ** *p* < 0.01, *** *p* < 0.001; (**b**) The flavonoids biosynthesis and accumulation pathway of *P. cyrtonema*. The circle represents metabolite, and the purple circle indicates metabolite annotated by KEGG enrichment analysis. The box represents genes, and the red box represents the gene selected for joint analysis. The solid line indicates the metabolic reactions in only one step, the dash line presents more than one step of the metabolic reaction.

**Figure 6 molecules-29-02248-f006:**
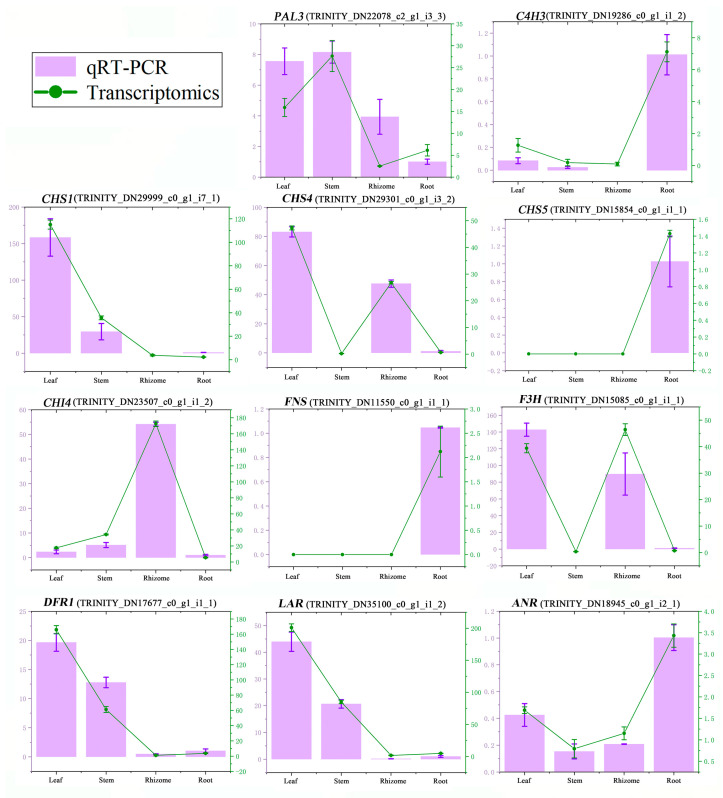
qRT-PCR validation of 11 DEGs. The *X*-axis means four tissues of *P. cyrtonema*. The left *Y*-axis represents the relative expression of each gene in four tissues by qRT-PCR, and the right *Y*-axis represents the FPKM value of each gene in four tissues by transcriptomics.

**Table 1 molecules-29-02248-t001:** Quantity and content of metabolites in different tissues of *P. cyrtonema*.

Tissues	Types of Metabolites	Number of Metabolites	Content of Metabolites (μg/g)
Leaf	Flavonols (11), Flavones (8), Flavanones (3), Dihydrochalcones (3), Isoflavones (1), Flavanol (1), Benzoic acid derivatives (9), Phenylpropanoids (6), Coumarins (4), Phenolic acids (2)	48	767.1181
Stem	Flavonols (8), Flavones (6), Flavanones (1), Dihydrochalcones (1), Anthocyanins (2), Isoflavones (1), Benzoic acid derivatives (7), Phenylpropanoids (6), Coumarins (3), Phenolic acids (2), Terpenoids (1)	38	550.0510
Rhizome	Flavonols (6), Flavones (8), Flavanones (2), Anthocyanins (1), Isoflavones 1), Benzoic acid derivatives (7), Phenylpropanoids (3)	28	1.7974
Root	Flavonols (9), Flavones (7), Flavanones (3), Dihydrochalcones (2), Anthocyanins (1), Isoflavones (2), Benzoic acid derivatives (9), Phenylpropanoids (4), Coumarins (1), Phenolic acids (2), Terpenoids (1)	41	9.2970

Note: The number of identified metabolites are represented by the numbers in parentheses.

**Table 2 molecules-29-02248-t002:** Differential flavonoid metabolites enriched by KEGG.

Metabolites	Class	KEGG ID	ID Annotation	Annotation
Astragalin	Flavonols	C12249	ath00944	Flavone and flavonol biosynthesis
Rutin	Flavonols	C05625	ath00944	Flavone and flavonol biosynthesis
Kaempferol	Flavonols	C05903	ath00941; ath00944	Flavonoid biosynthesis; Flavone and flavonol biosynthesis
Quercitrin	Flavonols	C01750	ath00944	Flavone and flavonol biosynthesis
Myricetin	Flavonols	C10107	ath00941; ath00944	Flavonoid biosynthesis; Flavone and flavonol biosynthesis
Quercetin	Flavones	C00389	ath00941; ath00944	Flavonoid biosynthesis; Flavone and flavonol biosynthesis
Luteolin	Flavones	C01514	ath00941; ath00944	Flavonoid biosynthesis; Flavone and flavonol biosynthesis
Cosmosiin	Flavones	C04608	ath00944	Flavone and flavonol biosynthesis
Naringenin	Flavanones	C00509	ath00941	Flavonoid biosynthesis
Naringin	Flavanones	C09789	ath00941	Flavonoid biosynthesis
Phlorizin	Dihydrochalcones	C01604	ath00941	Flavonoid biosynthesis
Phloretin	Dihydrochalcones	C00774	ath00941	Flavonoid biosynthesis
Catechin	Flavanols	C06562	ath00941	Flavonoid biosynthesis
4-Hydroxycinnamic acid	Phenylpropanoids	C00811	ath00940; ath00130; ath00998; ath00950; ath00350	Phenylpropanoid biosynthesis; Ubiquinone and other terpenoid-quinone biosynthesis; Biosynthesis of various secondary metabolites-part 2; Isoquinoline alkaloid biosynthesis; Tyrosine metabolism
Chlorogenic acid	Phenylpropanoids	C00852	ath00941; ath00940; ath00945	Flavonoid biosynthesis; Phenylpropanoid biosynthesis; Stilbenoid, diarylheptanoid and gingerol biosynthesis
Coniferaldehyde	Phenylpropanoids	C02666	ath00940	Phenylpropanoid biosynthesis
Ferulic acid	Phenylpropanoids	C01494	ath00940	Phenylpropanoid biosynthesis
Sinapic acid	Phenylpropanoids	C00482	ath00940	Phenylpropanoid biosynthesis
Umbelliferone	Coumarins	C09315	ath00940	Phenylpropanoid biosynthesis

**Table 3 molecules-29-02248-t003:** Genes involved in the synthetic pathway of flavonoids.

Number	Abbreviation	Name	Unigene Quantity	EC	Pathway	Source
1	—	Peroxidase	60	1.11.1.7	Ko00940	*P. kingianum* [41]
2	*Bgl*	Beta-glucosidase	34	3.2.1.21	Ko00940	*P. kingianum* [41]
3	*katG*	Catalase-peroxidase	21	1.11.1.21	Ko00940	
4	*TOGT1*	Scopoletin glucosyltransferase	16	2.4.1.128	Ko00940	
5	*HCT*	Shikimate *O*-hydroxycinnamoyltransferase	12	2.3.1.133	Ko00940 Ko00941	*P. kingianum* [41] *P. cyrtonema* [36]
6	*CAD*	Cinnamyl-alcohol dehydrogenase	11	1.1.1.195	Ko00940	*P. kingianum* [41]
7	*4CL*	4-coumarate-CoA ligase	10	6.2.1.12	Ko00940	*P. kingianum* [41] *P. cyrtonema* [36]
8	*CCOMT*	Caffeoyl-CoA *O*-methyltransferase	5	2.1.1.104	Ko00940 Ko00941	*P. kingianum* [37,41] *P. cyrtonema* [36]
9	*CYP73A(C4H)*	Trans-cinnamate 4-monooxygenase	5	1.14.14.91	Ko00940 Ko00941	*P. kingianum* [37,41] *P. cyrtonema* [36]
10	*CCR*	Cinnamoyl-CoA reductase	4	1.2.1.44	Ko00940	*P. kingianum* [41]
11	*PAL*	Phenylalanine ammonia-lyase	4	4.3.1.24	Ko00940	*P. kingianum* [41] *P. cyrtonema* [36]
12	*REF1*	Coniferyl-aldehyde dehydrogenase	4	1.2.1.68	Ko00940	*P. kingianum* [41]
13	*COMT*	Caffeic acid 3-*O*-methyltransferase	4	2.1.1.68	Ko00940	*P. kingianum* [41] *P. cyrtonema* [36]
14	*C3′H*	5-*O*-(4-coumaroyl)-d-quinate 3′-monooxygenase	2	1.14.14.96	Ko00940 Ko00941	*P. kingianum* [37] *P. cyrtonema* [36]
15	*CSE*	Caffeoylshikimate esterase	1	3.1.1.-	Ko00940	
16	*PRDX6*	Peroxiredoxin 6, 1-Cys peroxiredoxin	1	1.11.1.7	Ko00940	
17	*CHI*	Chalcone isomerase	4	5.5.1.6	Ko00941	*P. kingianum* [37,41] *P. cyrtonema* [36]
18	*PGT1*	Phlorizin synthase	4	2.4.1.357	Ko00941	
19	*FLS*	Flavonol synthase	2	1.14.20.6	Ko00941	*P. kingianum* [37] *P. cyrtonema* [36]
20	*F3′H*	Flavonoid 3′-monooxygenase	2	1.14.14.82	Ko00941 Ko00944	*P. kingianum* [37] *P. cyrtonema* [36]
21	*CHS*	Chalcone synthase	6	2.3.1.74	Ko00941	*P. kingianum* [37,41] *P. cyrtonema* [36]
22	*ANR*	Anthocyanidin reductase	1	1.3.1.77	Ko00941	*P. kingianum* [37]
23	*FNS*	Flavone synthase II	1	1.14.19.76	Ko00941	
24	*F3H*	Naringenin 3-dioxygenase	1	1.14.11.9	Ko00941	*P. cyrtonema* [36]
25	*DFR*	Bifunctional dihydroflavonol 4-reductase	2	1.1.1.219	Ko00941	*P. kingianum* [37] *P. cyrtonema* [36]
26	*LAR*	Leucoanthocyanidin reductase	1	1.17.1.3	Ko00941	*P. kingianum* [37] *P. cyrtonema* [36]
27	*UGT73C6*	Flavonol-3-*O*-l-rhamnoside-7-*O*-glucosyltransferase	3	2.4.1.-	Ko00944	*P. cyrtonema* [37]
28	*FG2*	Flavonol-3-*O*-glucoside l-rhamnosyltransferase	1	2.4.1.159	Ko00944	

**Table 4 molecules-29-02248-t004:** Primer sequence information of qRT-PCR.

**Gene Name**	**Forward** **Primer Sequence (5′-3′)**	**Reverse** **Primer Sequence (5′-3′)**
*18srRNA*	CGAGTCTATAGCCTTGGCCG	ATCCGAACACTTCACCGGAC
*PAL3*	AACGGAAATGGAGTGCACGG	GATATCTTCAGATCCGCTCCCC
*C4H2*	ACCCTCGAGTCCAAGAAAGG	CTCAGCCTCTTCCAGGTTCA
*CHS1*	GTAGGCCTGACTTTCCACCT	CCTCCACCTGGTCCAGTATC
*CHS4*	ATCGAACATACCGGAGGCAT	TTCCTCAGCACTTGTCTCGT
*CHS5*	CCGCTAAGGATCTTGCTGAG	CAATGGACGCTCGATAGCAA
*CHI1*	CAGTCTCAGCAACCAAGCTC	GATGCCGATGGCAGTAAAGG
*F3H*	ACTGCACCAGAGCTAGTGTT	ACACGTTGTAGGCCACCTTA
*DFR1*	CACCGATCCTGAGAACGAGA	CTCCAGCAGCTCTCATCGTA
*LAR*	GTCCAGGGTCTTGTTACGGA	GCCTTGTCGATGTCATGTCC
*ANR*	TGCTCAAGAAGGGCTATGCT	ACTGGTGTAGCGACATGGAA
*FNS*	GTGTGATTTTGGACTTTTT	CGTCTTCTATTTCTTGTTG

## Data Availability

The data presented in this study are available within this article and Appendix A. Raw Illumina sequencing reads have been deposited in the NCBI Sequence Read Archive Database under accession PRJNA1068333.

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
