# Peer review of "Integrated Metabolomics and Transcriptomics Analysis of Flavonoid Biosynthesis Pathway in Polygonatum cyrtonema Hua"

_molecules, 2024, doi:10.3390/molecules29102248_

Round 1
Reviewer 1 Report (Previous Reviewer 2)
Comments and Suggestions for Authors
The authors satisfactorily responded to all suggestions. However, the Figures in the manuscript need significant improvement related to their resolutions.
Author Response
Re: The authors satisfactorily responded to all suggestions. However, the Figures in the manuscript need significant improvement related to their resolutions.
Au: Thanks for your kind comments and sorry for low resolution figures. The resolutions of all the figures were improved.
Thanks again for your kindly suggestion.
Reviewer 2 Report (Previous Reviewer 3)
Comments and Suggestions for Authors
The article has been improved, however, I suggest some improvements in the resolution of the figures.
Figure 1, 2, 3 and 5 (a). Improve resolution.
Author Response
Re: The article has been improved, however, I suggest some improvements in the resolution of the figures.
Figure 1, 2, 3 and 5 (a). Improve resolution.
Au: Thanks for your kindly suggestion. We were sorry for low resolution figures. We improved the resolutions of all figures.
In addition, thanks again for your valuable improvements.
Reviewer 3 Report (New Reviewer)
Comments and Suggestions for Authors
In this paper, the authors present an integrated analysis combining metabolomics and transcriptomics to explore the flavonoid biosynthesis pathways in various tissues of the medicinal plant Polygonatum cyrtonema. The study utilized UPLC-MS/MS for a widely targeted analysis of phenolic compounds and RNA-seq to profile gene expression related to flavonoid biosynthesis. By correlating metabolite and gene expression data, the authors identified key pathways and enzymatic players involved in the regulation of flavonoid metabolism, providing insights into the tissue-specific accumulation of these compounds. Overall, this manuscript is a valuable contribution to the field of plant biochemistry and molecular biology, particularly regarding the biosynthesis of pharmacologically important flavonoids in Polygonatum cyrtonema. The study is methodologically sound and presents novel findings that enhance our understanding of plant secondary metabolism. Recommended minor comments primarily aim to provide additional context, detail, and clarity, which would make the paper stronger and suitable for publication in this journal.
Minor Comments
Introduction:
Expand the introduction to include background on flavonoid biosynthesis in medicinal plants to better situate the study within existing literature. Emphasize the pharmacological significance of flavonoids and the importance of understanding their biosynthetic pathways. Include recent findings from similar studies to position this research within the current scientific landscape.Clearly articulate the specific knowledge gaps this study aims to address. Highlight the unique aspects of Polygonatum cyrtonema Hua that make it a valuable subject for this type of analysis. Define the objectives more precisely to guide the reader on what the study specifically seeks to uncover about the flavonoid biosynthesis pathway in this plant. The authors should ensure that the text color for all the words in the manuscript is uniform, preferably black, to maintain consistency and readability throughout the document.
Methods:
The authors should provide detailed explanations of the methods used to identify the metabolites. It would be beneficial to specify whether the identification was based solely on database comparisons for putative identification or if it was confirmed by comparison with authentic standards. This information is crucial for validating the accuracy and reliability of the metabolomic analysis.Include more comprehensive details about the statistical approaches used for analyzing the differential accumulation of metabolites and gene expressions. Provide more details about the conditions under which the plant materials were collected (e.g., time of day, weather conditions), as these factors can influence metabolite profiles. Include information on the number of biological and technical replicates used in the experiments. This will help in assessing the robustness and reproducibility of the results. Elaborate on the statistical tests used to analyze the data. Specify which tests were used for which data types and why, including any corrections for multiple comparisons if applicable.
Figures and Tables:
Ensure all figures and tables are clearly labeled and include detailed legends. Increase the resolution of figures to improve clarity. Specifically, the heatmap in Figure 4 should avoid using a red-green color combination; instead, use color-blind-friendly colors.
Discussion:
Broaden the discussion to include comparisons with flavonoid biosynthesis in other related medicinal plants, which could help highlight unique or particularly effective biosynthetic strategies in P. cyrtonema. Provide a deeper analysis of the implications of the findings. Discuss how the identified pathways and genes compare with those found in other studies of similar plants and what these similarities or differences might mean. Suggest specific future studies that could be conducted based on the results. For instance, functional studies to validate the roles of newly identified genes in flavonoid biosynthesis or comparative studies with other species could be valuable follow-ups.
Comments on the Quality of English LanguageThe overall English language quality of the manuscript is good but could benefit from minor revisions to enhance clarity and flow. Here are specific areas for improvement:
Some sentences are overly long and complex, making them difficult to follow. Consider breaking them into shorter sentences to improve readability.
There are occasional grammatical errors, such as improper verb tense usage and subject-verb agreement. For example, "a total of 65 active substances were identified" should be carefully reviewed to maintain consistency in verb tense throughout the document.
Ensure consistency in the formatting of scientific names, abbreviations, and terminology throughout the manuscript. For example, "P. cyrtonema" is sometimes written in full and other times abbreviated. Choose one format and stick to it throughout the document.
Author Response
In this paper, the authors present an integrated analysis combining metabolomics and transcriptomics to explore the flavonoid biosynthesis pathways in various tissues of the medicinal plant Polygonatum cyrtonema. The study utilized UPLC-MS/MS for a widely targeted analysis of phenolic compounds and RNA-seq to profile gene expression related to flavonoid biosynthesis. By correlating metabolite and gene expression data, the authors identified key pathways and enzymatic players involved in the regulation of flavonoid metabolism, providing insights into the tissue-specific accumulation of these compounds. Overall, this manuscript is a valuable contribution to the field of plant biochemistry and molecular biology, particularly regarding the biosynthesis of pharmacologically important flavonoids in Polygonatum cyrtonema. The study is methodologically sound and presents novel findings that enhance our understanding of plant secondary metabolism. Recommended minor comments primarily aim to provide additional context, detail, and clarity, which would make the paper stronger and suitable for publication in this journal.
Minor Comments
Re: Introduction:
Expand the introduction to include background on flavonoid biosynthesis in medicinal plants to better situate the study within existing literature. Emphasize the pharmacological significance of flavonoids and the importance of understanding their biosynthetic pathways. Include recent findings from similar studies to position this research within the current scientific landscape. Clearly articulate the specific knowledge gaps this study aims to address. Highlight the unique aspects of Polygonatum cyrtonema Hua that make it a valuable subject for this type of analysis. Define the objectives more precisely to guide the reader on what the study specifically seeks to uncover about the flavonoid biosynthesis pathway in this plant. The authors should ensure that the text color for all the words in the manuscript is uniform, preferably black, to maintain consistency and readability throughout the document.
Au: Thanks for your helpful comments. We expanded the pharmacological effects of flavonoids in medicinal plants, as well as the differences in the effects of different components of flavonoids in Polygonatum (Lines 42-58). We added a description and analysis of the flavonoid synthesis pathway in medicinal plants, and clarified the reasons for studying the flavonoid synthesis pathway in P. cyrtonema (Lines 85-97).
Re: Methods:
The authors should provide detailed explanations of the methods used to identify the metabolites. It would be beneficial to specify whether the identification was based solely on database comparisons for putative identification or if it was confirmed by comparison with authentic standards. This information is crucial for validating the accuracy and reliability of the metabolomic analysis. Include more comprehensive details about the statistical approaches used for analyzing the differential accumulation of metabolites and gene expressions. Provide more details about the conditions under which the plant materials were collected (e.g., time of day, weather conditions), as these factors can influence metabolite profiles. Include information on the number of biological and technical replicates used in the experiments. This will help in assessing the robustness and reproducibility of the results. Elaborate on the statistical tests used to analyze the data. Specify which tests were used for which data types and why, including any corrections for multiple comparisons if applicable.
Au: Thanks for your suggestion. We provided explanations of the methods used to identify the metabolites (Lines 362-373). And the weather conditions and sampling time were described in Lines 345-347. A data analysis section was also added. Please see details in the Methods section.
Re: Figures and Tables:
Ensure all figures and tables are clearly labeled and include detailed legends. Increase the resolution of figures to improve clarity. Specifically, the heatmap in Figure 4 should avoid using a red-green color combination; instead, use color-blind-friendly colors.
Au: Thanks for your suggestion. We added a general title for Figure 5 (Line202). We redrawn Figure 4 and replaced the red-green color with other colors. We improved the resolutions of all figures.
Re: Discussion:
Broaden the discussion to include comparisons with flavonoid biosynthesis in other related medicinal plants, which could help highlight unique or particularly effective biosynthetic strategies in P. cyrtonema. Provide a deeper analysis of the implications of the findings. Discuss how the identified pathways and genes compare with those found in other studies of similar plants and what these similarities or differences might mean. Suggest specific future studies that could be conducted based on the results. For instance, functional studies to validate the roles of newly identified genes in flavonoid biosynthesis or comparative studies with other species could be valuable follow-ups.
Au: Thanks for your suggestion. We discussed the metabolic pathways enriched with flavonoid related genes in medicinal plants and the flavonoid related pathways reported in Polygonatum (Lines 254-267). And we suggest specific future studies that could be conducted based on the results (Lines 281-283).
Re: The overall English language quality of the manuscript is good but could benefit from minor revisions to enhance clarity and flow. Here are specific areas for improvement:
Some sentences are overly long and complex, making them difficult to follow. Consider breaking them into shorter sentences to improve readability.
There are occasional grammatical errors, such as improper verb tense usage and subject-verb agreement. For example, "a total of 65 active substances were identified" should be carefully reviewed to maintain consistency in verb tense throughout the document.
Ensure consistency in the formatting of scientific names, abbreviations, and terminology throughout the manuscript. For example, "P. cyrtonema" is sometimes written in full and other times abbreviated. Choose one format and stick to it throughout the document.
Au: Thanks for your helpful comments. We reviewed to maintain consistency in verb tense throughout the document. We chose the format "P. cyrtonema" and stick to it throughout the document.
Thank you very much for your valuable comments for further improvement of our manuscript. All change was highlighted with red color. Thanks again.
Reviewer 4 Report (New Reviewer)
Comments and Suggestions for Authors
please see attached file

The langauge requires some editing
Author Response
General comment: Many words at the end of the sentences are cut off. This makes reading difficult.
- L9. the word "which" is used incorrectly. This is a very common error throughout the
manuscript. For instance in this sentence it means that Polygonatum are phenolic compounds. That is not what the authors mean to state.
- L18. the word firstly is used incorrectly. It should be first study to identify. This error also appears in other parts of the manuscript.
- L36 it is not clear what Huang Jing refers to in this sentence.
- L37 on Polygonatum sibiricum Delar. ex Redoute, Polygonatum kingianum Coll. et Hemsl and Polygonatum cyrtonema Hua [3].
Once a genus name is written in full, the others can be abbreviated to P.
- L41 enhance should be enhances
- L53 which is used incorrectly
- L71 genes identification should be gene identification
- L77 excavated is a wrong word to use in this sentence
- L80 such pathway … . Should read such pathways or such a pathway
- L82-85 sentence is not clear and needs rephrasing
- L206 allows should be allow
- L210 no need to repeat authority name of the species (Hua) –it has already been stated before.
- L210 and 212 should be phenolic metabolites –not as written
- L216 under the forest should be in the forest
- L219 no need for respectively
- L 220 remove the word the
- L220/221 that have
- L225 – firstly used incorrectly again 19.L228 were to be replaced by are
- L239 which is used incorrectly
- 245 no need to repeat enzyme
- L246 were enriched
- L261 which is used incorrectly
- L269 please rephrase and check the grammar
- L285 please check the scientific name of Chayote
- L290 replace might with may 27.L294 correlated with
- L304 in the leaves
29 L304 replace while with and
Au: Thanks for your careful checks. We corrected these according to your suggestion. All change was highlighted with red color.
In addition, thank you very much for your careful reading of our manuscript. Thanks again for your positive evaluation and support for the MS.

This manuscript is a resubmission of an earlier submission. The following is a list of the peer review reports and author responses from that submission.
Round 1
Reviewer 1 Report
Comments and Suggestions for Authors
the resolution of all the figures in the manuscript needs to be improved before I can read and understand the results. So I would suggest to solve this first and resubmit again. The authors should also read the manuscript carefully as the figure 4 in section 2.7 should be figure 6.
Comments on the Quality of English LanguageModerate editing of English language required
Author Response
Re: The resolution of all the figures in the manuscript needs to be improved before I can read and understand the results. So I would suggest to solve this first and resubmit again. The authors should also read the manuscript carefully as the figure 4 in section 2.7 should be figure 6.
Au: Thanks for your kind comments and sorry for low resolution figures and wrong figure number. Yes, ”figure 4” should be “figure 6”(Line 207). The resolutions of all the figures were improved.
In addtion, we rewrote the Introduction section and added the conclusion section. We re-examined the MS carefully and highlighted all change with red color. Thanks again.
Reviewer 2 Report
Comments and Suggestions for Authors
Dear Editor and Authors, I recommend the publication of this manuscript after minor revision.
The following changes are recommended and some clarifications should be made:
Modification in the section Abstract
Pg. 1, Line 9-11: The first sentence of the Abstract should be rephrased and clearly presented. Please use "are", instead of "were".
Pg. 1, Line 14-16: The full name of tested species Polygonatum should be provided.
Pg. 1, Line 18, 23: The full names of DAMs and DEGs should be provided.
Pg. 1, Line 19: "coumarins" should be in singular.
Modification in the section Introduction
Pg. 2, 87: The genus names of D. loddigesii should be provided.
The Introduction section could be enriched with more specific data for metabolomics and transcriptomics in various species of Polygonatum.
Modification in the section Results
Table 1: The content of metabolites could be expressed in µg/g or eventually in mg/g. In the note of Table 1, the term "amounts" should be replaced with "number of identified".
Figures: All Figures in the manuscript are blurred. Please, provide clear pictures for all Figures.
Modification in the section Discussion
Pg. 9, Line 206-216: The following part could be excluded, since it is too general and not so important for the present study. "The metabolomics technology was first proposed by Professor Nicholson in 1999, that is, taking small molecular metabolites in organisms as the research object, and conducting comprehensive qualitative and quantitative analysis through a variety of modern analytical techniques [34]. With the development of metabolomics technology, extensive targeted metabolomics techniques of UPLC-ESI-MS/MS were widely used to identify metabolites of Chinese medicine, which has the advantages of high sensitivity, low cost and high throughput compared with traditional chromatography [35, 36]. The applications of metabolomics in medicinal plants include identification and quality evaluation of Chinese herbal medicines [37], clarification mechanism of Chinese herbal medicine preparation [38], as well as identification of metabolites types and contents [39], elucidation of metabolic pathway and regulatory network [40]."
The finding that leaves were the richest source of flavonoids was compare with many studies, but potential explanation for these results is missing. Any clarification for these data?
Modification in the section Materials and Methods
The identification of compounds by UPLC-ESI-MS/MS analysis should be presented.
Author Response
Dear Editor and Authors, I recommend the publication of this manuscript after minor revision. The following changes are recommended and some clarifications should be made:
Re: Modification in the section Abstract
The first sentence of the Abstract should be rephrased and clearly presented. Please use "are", instead of "were". The full name of tested species Polygonatum should be provided. The full names of DAMs and DEGs should be provided. "coumarins" should be in singular.
Au: Thanks for your kindly suggestion. We reorganized the first sentence of the Abstract. The full name of Polygonatum, DAMs and DEGs were provided. All corrections in our resubmitted manuscript marked in red.
Re: Modification in the section Introduction
The genus names of D. loddigesii should be provided. The Introduction section could be enriched with more specific data for metabolomics and transcriptomics in various species of Polygonatum.
Au: Thanks for your suggestion. We corrected the D. loddigesii into Dendrobium loddigesii Rolfe (Line 96). We rewrote the section of introduction according to the suggestion. The second paragraph mainly described the identification of Polygonatum flavonoids. The third paragraph focused on the enzymes related to the flavonoid synthesis pathway in Polygonatum through transcriptomics. Please see details in the Introduction section.
Re: Modification in the section Results
Table 1: The content of metabolites could be expressed in µg/g or eventually in mg/g. In the note of Table 1, the term "amounts" should be replaced with "number of identified". Figures: All Figures in the manuscript are blurred. Please, provide clear pictures for all Figures.
Au: Thanks for your suggestion. We corrected the “ng/g” into “µg/g” to express the content of metabolites, and the “number of identified” into “amounts” (Lines117-118). We improved the resolutions of all figures.
Re: Modification in the section Discussion
The following part could be excluded, since it is too general and not so important for the present study. "The metabolomics technology was first proposed by Professor Nicholson in 1999, that is, taking small molecular metabolites in organisms as the research object, and conducting comprehensive qualitative and quantitative analysis through a variety of modern analytical techniques [34]. With the development of metabolomics technology, extensive targeted metabolomics techniques of UPLC-ESI-MS/MS were widely used to identify metabolites of Chinese medicine, which has the advantages of high sensitivity, low cost and high throughput compared with traditional chromatography [35, 36]. The applications of metabolomics in medicinal plants include identification and quality evaluation of Chinese herbal medicines [37], clarification mechanism of Chinese herbal medicine preparation [38], as well as identification of metabolites types and contents [39], elucidation of metabolic pathway and regulatory network [40]."
Au: Thanks for your suggestion. This part was deleted. And we reorganized the first paragraph of the section Discussion. Please see details in Lines 211-213.
Re: The finding that leaves were the richest source of flavonoids was compare with many studies, but potential explanation for these results is missing. Any clarification for these data?
Au: A good suggestion! We disscussed potential reasons about high flavonoids content in Polygonatum leaves (Lines 224-228).
Re: Modification in the section Materials and Methods
The identification of compounds by UPLC-ESI-MS/MS analysis should be presented.
Au: Thanks for your suggestion. Lines 349-353 was added to explain the identification of compounds by UPLC-ESI-MS/MS.
Thank you very much for your valuable comments for further improvement of our manuscript. All change was highlighted with red color. Thanks again for your positive evaluation and support for the MS.
Reviewer 3 Report
Comments and Suggestions for Authors
Integrated Metabolomics and Transcriptomics Analysis of Flavonoid Biosynthesis Pathway in Polygonatum cyrtonema Hua
Abstract:
The methodology needs to be better explained in this topic;
Introduction
It enters the lung, spleen and kidney meridians, and has the effects of nourishing qi and yin, invigorating the spleen, benefiting the kidney, and moistening the lung. How? Explain?
The second paragraph and the third paragraph are not consistent because the information is not interconnected.
D.loddigesii . What genus? Insert in full.
Results
Figure 1. Needs to be improved resolution
Figure 2. Needs to be improved resolution
Figure 3. Needs to be improved resolution
Figure 4. Needs to be improved resolution
Figure 5. Needs to be improved resolution
There are two nominees as Figure 4
Discussion
This topic needs improvement, as the discussion starts with an introduction, which I consider not relevant
What is the conclusion of this research? What impacts are considered relevant and significant?
Author Response
Integrated Metabolomics and Transcriptomics Analysis of Flavonoid Biosynthesis Pathway in Polygonatum cyrtonema Hua
Re: -Abstract
The methodology needs to be better explained in this topic;
Au: Thanks for your kindly suggestion. We corrected it according to your suggestion (Lines15-18).
Re: -Introduction
It enters the lung, spleen and kidney meridians, and has the effects of nourishing qi and yin, invigorating the spleen, benefiting the kidney, and moistening the lung. How? Explain?
Au: Thanks for your careful reading. “It enters the lung, spleen and kidney meridians, and has the effects of nourishing qi and yin, invigorating the spleen, benefiting the kidney, and moistening the lung” was a description of the channel tropism and function of P. cyrtonema in the Pharmacopoeia. Channel tropism refers therapeutic action of drugs to a certain parts of the body. For a clearer description, We corrected it and described its function according to modern pharmacological effects (Lines 41-43)
Re: The second paragraph and the third paragraph are not consistent because the information is not interconnected.
Au: Thanks for your suggestion. We rewrote the section of introduction according to your suggestion. The second paragraph mainly described the identification of Polygonatum flavonoids. The third paragraph focused on the enzymes related to the flavonoid synthesis pathway in Polygonatum through transcriptomics. Please see details in the Introduction section.
Re: D.loddigesii . What genus? Insert in full.
Au: Thanks for your careful checks. The genus names of D. loddigesii was Dendrobium loddigesii Rolfe. We corrected it (Line 96).
Re: -Results
Figure 1. Needs to be improved resolution. Figure 2. Needs to be improved resolution. Figure 3. Needs to be improved resolution. Figure 4. Needs to be improved resolution. Figure 5. Needs to be improved resolution. There are two nominees as Figure 4
Au: Thanks for your kind comments and sorry for low resolution figures and wrong figure number. Yes, ”figure 4” should be “figure 6”(Line 207). The resolutions of all the figures were improved.
Re: -Discussion
This topic needs improvement, as the discussion starts with an introduction, which I consider not relevant
Au: Thanks for your helpful comment. The first part of the discussion was deleted and rewrote. Please see details in Lines 211-213.
Re: What is the conclusion of this research? What impacts are considered relevant and significant?
Au: We are grateful for your advice. Our conclusion was that metobolites and genes associated with flavonoids biosynthesis pathway were identified. Different flavonoid metabolites were more likely to be regulated by 4CL1, 4CL2, CHS2, CHS3, DFR1, LAR genes. Our results provide valuable information on Polygonatum flavonoid regulation. Please see details in Lines 318-327 .
Thank you very much for your valuable comments for further improvement of our manuscript. Thanks again.